# Red Beetroot’s NMR-Based Metabolomics: Phytochemical Profile Related to Development Time and Production Year

**DOI:** 10.3390/foods10081887

**Published:** 2021-08-15

**Authors:** Ottavia Giampaoli, Fabio Sciubba, Giorgia Conta, Giorgio Capuani, Alberta Tomassini, Giorgio Giorgi, Elisa Brasili, Walter Aureli, Alfredo Miccheli

**Affiliations:** 1Department of Chemistry, Sapienza University of Rome, P.le Aldo Moro 5, 00185 Rome, Italy; ottavia.giampaoli@uniroma1.it (O.G.); giorgia.conta@uniroma1.it (G.C.); giorgio.capuani@uniroma1.it (G.C.); alberta.tomassini@uniroma1.it (A.T.); 2NMR-Based Metabolomics Laboratory (NMLab), Sapienza University of Rome, P.le Aldo Moro 5, 00185 Rome, Italy; alfredo.miccheli@uniroma1.it; 3R&D, Aureli Mario S.S. Agricola, Via Mario Aureli 7, 67050 Ortucchio (Aq), Italy; r.d@aurelimario.com (G.G.); produzione@aurelimario.com (W.A.); 4Department of Environmental Biology, Sapienza University of Rome, P.le Aldo Moro 5, 00185 Rome, Italy; elisa.brasili@uniroma1.it

**Keywords:** red beetroot, seasonality, harvest time, NMR-based metabolomics, inorganic nitrates, dopamine, betanin

## Abstract

Red beetroot (RB) is a well-known health-promoting food consumed worldwide. RB is commonly used in food processing and manufacturing thanks to the high content of components that can also be employed as natural coloring agents. These bioactive molecules vary their concentration depending on beetroot seasonality, harvest time and climate conditions. The first objective of this study was to evaluate the variation of the RB phytochemical profile related to the root development during three different harvest times, using an ^1^H-NMR-based metabolomic approach. Changes of carbohydrates and secondary metabolite concentrations were observed from July to September. Secondly, we compared the metabolic profiles of the final processed beet juices in three different production years to observe the effect of climate conditions on the RB’s final product metabotype. A PCA analysis performed on juice extracts showed that production years 2016 and 2017 were characterized by a high content of choline and betaine, while 2018 by a high content of amino acids and dopamine and a low content of inorganic nitrates. This study suggests that the harvest time and roots growth conditions could be used to modulate the RB phytochemical profile, according to the final requirements of use, food or coloring agent source.

## 1. Introduction

*Beta vulgaris* subsp. *vulgaris* var. *rubra*, commonly known as red beetroot (RB), is an herbaceous biennial taproot belonging to the *Chenopodiaceae* family. Currently, RB represents an important market vegetable crop which has a worldwide distribution [1]. RB is cultivated in temperate climate zones and prefers slightly acidic and clayey soils, despite adapting well to any type of soil. RB can be seeded across the whole year, even though is preferentially seeded between March and June in order to avoid low temperatures [2]. The growth period is between 75 and 90 days in summer, but it can increase up to 120 days or even longer in winter [2]. RBs can be consumed raw or cooked and are also largely employed in the processing industry to produce juices. In recent years, RB has attracted much attention as a health-promoting food because of the high content of nitrates, molecular species that must be introduced in the organism through the diet. Supplementation with RB juice has been shown to improve exercise performance and cardiovascular responses, thanks to the high content of antioxidant compounds and the increased availability of nitric oxide (NO) [3,4,5,6]. Inorganic nitrates (NO^−^_3_) undergo a reduction to nitrites (NO^−^_2_) by oral bacteria and then to NO in the gastric acid milieu through a non-enzymatic reaction [7]. NO can diffuse in tissues even faster than O_2_ and can easily reach capillaries where it acts as vasodilatory agent, improving the human skeletal muscle function and acting on the mechanism of muscle relaxation [8]. These properties lead to an increase in blood flow and oxygen supply, particularly effective in local hypoxic states such as physical activity and cardiovascular diseases [9]. Nitrate and nitrite intake could be potentially associated with the development of nitrosamines, compounds considered as probable carcinogens for humans by the International Agency for Research on Cancer (IARC) [10,11]. Nonetheless, the significant amounts of antioxidants in RBs inhibit this process, making them ideal for consumption as functional foods [5,12,13,14]. The most important antioxidants in RBs are betalains which are promising compounds against diseases involving oxidative stresses [15,16]. These compounds are natural colorants largely employed by the food industry [17,18]. The observed beneficial effects of RBs are relatable to their phytochemical profile, that in turn depends on agronomic and pedoclimatic conditions [19,20,21,22]. Indeed, an high biochemical variability, related to cultivar specificity and geographical origin, has been shown in previous studies [23,24,25], but yet the variability of the RB chemical composition, induced by harvest time and seasonality, has not been deeply investigated. Therefore, the scarce knowledge of a defined metabolic profile of red beetroot could barely help the producer to address its use in terms of a top-quality food for health care. In this regard, the present study aims at defining the metabolic profile of RB and its evolution during the root development, as well as comparing beet juices from different years (2016, 2017 and 2018) to evaluate the influence of the climatic conditions on the phytochemical profile.

## 2. Material and Methods

### 2.1. Plant Materials

RBs (*Beta vulgaris* subsp. *vulgaris* var. *rubra* cv Pablo) were cultivated by Aureli Mario S.S. Agricola (Ortucchio, Aq, Italy). The fields are located in the Fucino plan (Aq, Italy) with a mean altitude of 680 m above sea level at 41°52′ N latitude and 12°12′ E longitude. The beets were grown in the same environmental conditions and cultivation methods according to organic practice. For the root development study, beets were seeded in May 2018 and 28 redroots were harvested in the months of July, August and September, corresponding, respectively, to third, fourth and fifth months of development. Commercial RB juices were provided by Aureli Mario S.S. Agricola and were processed in their industrial plants from beets collected in the Fucino fields. Bottles from the years 2016 (4 samples), 2017 (5 samples) and 2018 (6 samples) were provided. All the juice samples were manufactured from roots harvested in September.

### 2.2. Sample Preparation

A total of 1.5 g of grated RBs for each harvest time and 1.5 mL of juice were extracted following a modified Bligh–Dyer protocol [21]. More details can be found in Supporting Information (Extended Materials and Methods).

### 2.3. Nitrate Determination

Nitrate content in RB juices was measured employing an RQflex^®^ 20 Reflectoquant^®^ reflectometer (Merck KGaA, 64271 Darmstadt, Germany) according to the kit instructions.

### 2.4. NMR Experiments

All spectra were recorded at 298 K on a Bruker AVANCE III spectrometer (Bruker BioSpin, Karlsruhe, Germany), equipped with a Bruker multinuclear z-gradient inverse probe head operating at the proton frequency of 400.13 MHz. More details can be found in Supporting Information (Extended Materials and Methods). 

### 2.5. Statistical Analysis

Multivariate PCA was performed on the data matrix with the Unscrambler ver. 10.5 software (Camo Software AS, Oslo, Norway) and univariate one-way ANOVA was performed with SigmaPlot 14.0 software (Systat Software Inc., San Jose, CA, USA). More details can be found in Supporting Information (Extended Materials and Methods). 

## 3. Results

A representative spectrum of RB hydroalcoholic extract is reported in Appendix A. A total of 36 metabolites were identified and 31 were quantified from ^1^H NMR spectra of the hydroalcoholic extract (Appendix A). A resonance assignment was carried out on the basis of the signal chemical shift, multiplicity, TOCSY, HSQC and HMBC correlations. The ^1^H chemical shifts, multiplicity and the ^13^C chemical shifts of the identified molecules are reported in Appendix A. 

### 3.1. Metabolic Variations Related to Root Development 

In order to evaluate the metabolic changes occurring during RB development, a non-supervised PCA analysis was carried out on roots harvested in July, August and September 2018. 

The first four components of the PCA model explained 80% of the overall variance, with the first (PC1) accounting for 38% and the second (PC2) for 19%. A spontaneous grouping according to the harvest time was identified along the PC2 axis in the PCA scores plot (Figure 1), with July roots placed at lower PC2 values and the September ones at high values. A statistically significant difference (*p* < 0.05) was found on each RB’s groups according to the ANOVA analysis, conducted on the PC2 values of RB samples.

In Figure 2, the normalized PC2 loading values of the metabolites were reported. Variables with normalized loading values greater than 0.39 and lower than −0.39 were considered significant for the model (*p* < 0.05) according to the Pearson table of critical values for correlations. 

PC2 was directly proportional to glucose and myo-inositol, while inversely proportional to dopamine, glutamine, malonate, fructose, aspartate, hydroxybenzoate, citrate, GABA, betanin, choline and sucrose. The concentrations of the significant metabolites resulting important for the PCA model were reported in Table 1 and, since the main focus of this analysis was the temporal evolution of the phytochemical composition of RBs, the changes were monitored from July to August and from August to September.

The results showed a linear increase in glucose and myo-inositol levels from July to September. On the contrary, fructose decreased from July to August and increased in September while sucrose, aspartate, choline and betanin linearly decreased from July to September. Dopamine decreased from July to August, reaching a plateau in September.

### 3.2. Metabolic Variations Related to Seasonal Changes

A non-supervised PCA analysis was performed on juice extracts produced in 2016, 2017 and 2018, in order to evaluate the influence of the climatic changes occurring between years on the metabolic profile of RB commercial juices. The analysis provided a model whose first two components accounted for 90% of the overall variance, with PC1 accounting for 63% and PC2 for 27%. The scores plot (Figure 3) showed a spontaneous grouping along the PC1 axis according to the year, with 2017 samples at highest values of PC1 and juices from 2018 at lowest values. In the PC2 axis, we could identify 2016 samples of juices at lowest values, whereas 2017 and 2018 at the highest. The groupings were statistically significant (*p* < 0.05) according to the ANOVA analysis conducted on PC1 values of the samples.

The normalized PCA loading plot of the first component shows which variables were important along with the PC2 factor (Figure 4). Depending on the Pearson table of critical values for correlation, variables with normalized loading values greater than 0.53 and lower than −0.53 were considered significant for the model (*p* < 0.05). 

The variables proportional to low values of PC1 were trigonelline, dopamine, tyrosine, malonate, phenylalanine, asparagine, glutamine, isoleucine, GABA, valine, threonine, leucine, dimethylamine, fumarate, sucrose, malate, alanine, acetamide, betanin and fumarate, while for high values of PC1 the important variables were choline and citrate.

The metabolites significant for the separation of RB juices according to the year of production were reported in Table 2.

The results showed that, comparing the production of the year 2017 to years 2016 and 2018, a general decrease of all measured molecules was observed, excepted alanine, dimethylamine, betaine and choline. Choline was the only molecule to be more abundant in 2017 than 2016 and 2018. From the comparison of juices from 2016 to 2017, an increase of threonine, glutamine, asparagine, tyrosine, phenylalanine, malate, fumarate, dimethylamine and dopamine and a decrease in glutamate, fructose, glucose, formate, choline and betanin were observed.

## 4. Discussion

The commercialization of agricultural products is currently facing an ever-increasing demand of quality safety and healthy nutrient contents by the consumers. In this regard, the recognition of a scientific-based optimal harvest time is of paramount importance for the agro-farm industry to ensure a uniform quality, particularly for the transformed products such as juices.

The chemical composition of the raw material is affected by factors such as different cultivars, ripening evolution, developmental stage, geographical origin, growing sites, seasonal and climatic differences [26].

In this context, a metabolomic approach represents a powerful method for providing a comprehensive profile of the biochemical composition starting from the raw material and up to the ready-to-consume product, as shown in two preceding papers on orange and purple carrots [19,21].

In the present study, the variation on the phytonutrient profiles of RBs during the root development and the different composition of ready-to-drink red beetroot juices of three different annual production were investigated by ^1^H NMR-based metabolic profiling.

### 4.1. Metabolic Variations Related to Root Development 

The only metabolites that showed an increase during root development were glucose and myoinositol, which were also significantly different in the univariate analysis with an abrupt increase from August to September.

The decrease of sucrose concentration could be ascribable to an increase in the RB fiber content, since sugar metabolism provides substrates for the biosynthesis of starch, cellulose, callose and proteins, and a similar trend has already been observed in the development of other taproots, i.e., onions [27]. Sucrose metabolism and its degrading products, glucose and fructose, have been involved in many developmental processes in plants and in the regulation of gene expression as well as in the response to abiotic stresses [28]. There are two enzymes that catalyze the cleavage of sucrose, namely, sucrose synthase and invertases, which are differently expressed during the root development. High levels of sucrose synthase activity have been observed in mature roots and tubers such as radish, carrot and potato [29,30].

It is important to underline that we focused on the development of the red beet root which represents the relevant agronomical part in relation to the economic value of the crop. However, the aerial part of the plant was also developed, and that could affect the sucrose content during the root growth. Indeed, a common traditional agronomic practice envisages the cut of the leaves just a few days before the harvest to prevent the decrease and to increase the sugar content of the root.

The observed decrease in sucrose and fructose, along with the increase in glucose concentrations, could also be related to the control of osmotic pressure as a response to the change of the climate conditions in summer. In agreement with this hypothesis, an increase in myo-inositol concentration was also observed. The first step of myo-inositol biosynthesis is the conversion of D-glucose-6-P, which is further metabolized into several inositol phosphates with many functional roles, e.g., the response to light, heat, drought and osmotic stresses [31]. Furthermore, the oxidation of free myo-inositol to D-glucuronic acid is involved to the biogenesis of pectin, hemicelluloses and related structures in plant cell walls [32]. Therefore, the observed concomitant increase in glucose and myo-inositol could be due to two factors: the growth and fiber content of the root at the end of the development, and the metabolic response to the increased heat and drought in August.

The concomitant decrease in dopamine and betanin levels could suggest a different regulation of the glycosylation of betalains synthesis and degradation at the transcriptional level during development. Betalains are water soluble nitrogenous pigments characterized by a common scaffold, betalamic acid, condensed with cyclo-Dopa or various amino acids. Betalains degradation depends on b-glucosidase activity that has been demonstrated both in *B. vulgaris* leaves and roots [33,34].

In agreement with the present data, a decrease in both the sugar and betanin content in red beet was observed according to the weigh and caliber growth during ripening [35]. 

It is interesting to highlight that the average weight of RBs harvested in August was of about 300 g, and it rose to 380 g in September, while the content of betanin went from 38 mg/g in August to 23 mg/g in September. By combining these two data, it is possible to determinate that an early harvest would provide an increase of about 23% in terms of bioactive molecules. The presented information suggests the use of a comprehensive phyto-profiling to establish an optimal harvest time in relation to the intended use of the raw material, namely, in the food or color industry, overcoming the traditional crop harvest parameters, which are based only on the possible maximum yield. 

### 4.2. Metabolic Variations Related to Seasonal Changes

Regarding the chemical differences among years of cultivation, one of the main aspects of this work was that the beets were cultivated in the same soil, with the same growing conditions and harvested at the same time. Therefore, all the observed differences were only ascribable to the climatic conditions. The analysis of the climatic reports of the Abruzzo region [36] showed that the year 2016 had the same average, minimum and maximum temperatures and rainfall of the typical Fucino climate. Therefore, the composition of the juice obtained from 2016 RBs can represent a geographic-specific standard. Year 2017 was exceptionally dry, with only a total 36.2 mm of precipitation from June to August (Table 3). 

Considering the differences between juices from 2016 to 2017, beets harvested in the drier year had lower amounts of all metabolites with the exception of osmolytes such as choline and betaine. We could hypothesize that drought impaired the taproot development, affecting juice composition. The relative increase in choline and betaine in 2017 juices compared to 2016 could be a plant response to this abiotic stress. 

The year 2018 was the coldest of the three considered years. Temperature was an important aspect for the plant growth, since RBs need a temperature above 11 °C to develop and germinate and many days were below this temperature in this year. 

The increased amount of some small nitrogen-containing molecules, namely, dopamine, dimethylamine, asparagine and glutamine, could be related to nitrogen metabolism and storage in plants at low temperatures. Furthermore, it is known that nitrate uptake from the soil is largely dependent on the air and soil temperature [37] and the data on the lower content of nitrate in 2018 juices in respect to 2016 and 2017 ones (Table 4), further supported this observation. 

The concomitant decrease in glutamate and betanin levels and the increase in glutamine and dopamine could be explained by their metabolism. In fact, glutamate is transformed to glutamine with the addition of an amino group, while dopamine is an intermediate in the synthesis pathway of betalains, and the biosynthesis of the latter could be reduced to increase the circulating amount of catechols. 

The increase in phenylalanine and tyrosine, which are known precursors of phenylpropanoid intermediates, could be a response to cold stress similar to what is reported in literature for Arabidopsis, cowpea and carrots [21,38,39,40]. In particular, the increase in threonine levels was rather interesting since it is known that this amino acid is involved in the plant response against abiotic stresses [41]. Even the non-uniform changes in the organic acid content, i.e., the decrease in citrate and the increase in the malate content, could be related to the plant response to a cold climate [42].

## 5. Conclusions

In conclusion, we showed how the comprehensive metabolic profile, obtained through NMR-based metabolomics, could represent a powerful tool to unravel the metabolic processes involved in the red beetroot development and in the year variability. The knowledge of the chemical composition may help the producer to optimize the harvest time in terms of important nutrient compounds to obtain a standardized top-quality transformed product and may help in contrasting unpleasant effects to abiotic climate changes through years by adopting appropriate agronomic interventions.

Furthermore, the knowledge of the precise biochemical composition is of paramount importance when evaluating the assumption of vegetable-derived products as health sustaining therapy as recently suggested for red beetroot juices for people in quarantine isolation for COVID-19 infection or in long COVID-affected patients [43].

## Figures and Tables

**Figure 1 foods-10-01887-f001:**
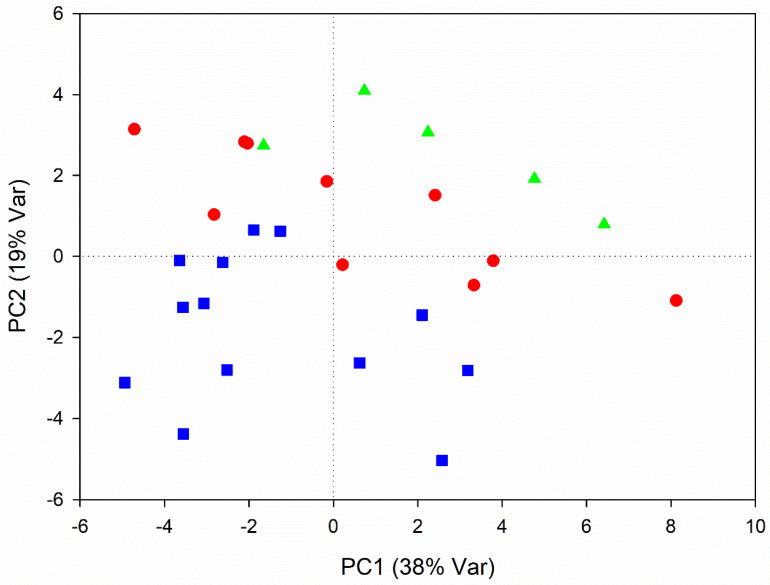
PCA score plot of RBs. Beets harvested in July are indicated in blue squares, August in red circles and September in green triangles.

**Figure 2 foods-10-01887-f002:**
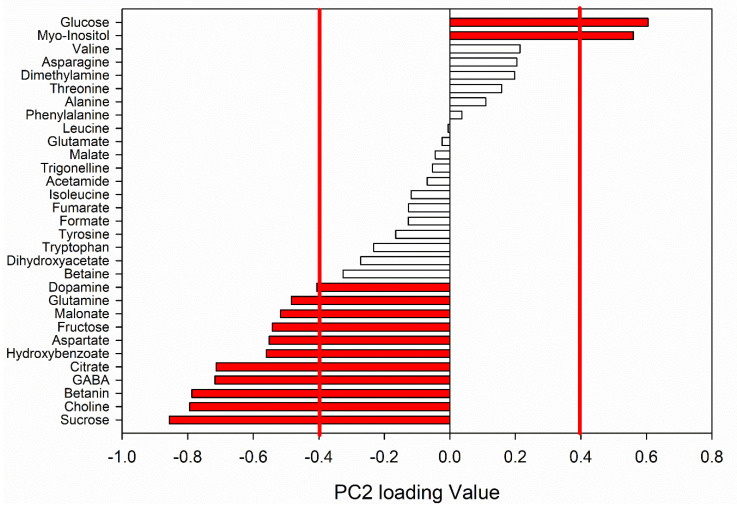
Normalized PC2 Loadings of RBs harvested in July, August and September. In red, the variables with *p* < 0.05 are evidenced.

**Figure 3 foods-10-01887-f003:**
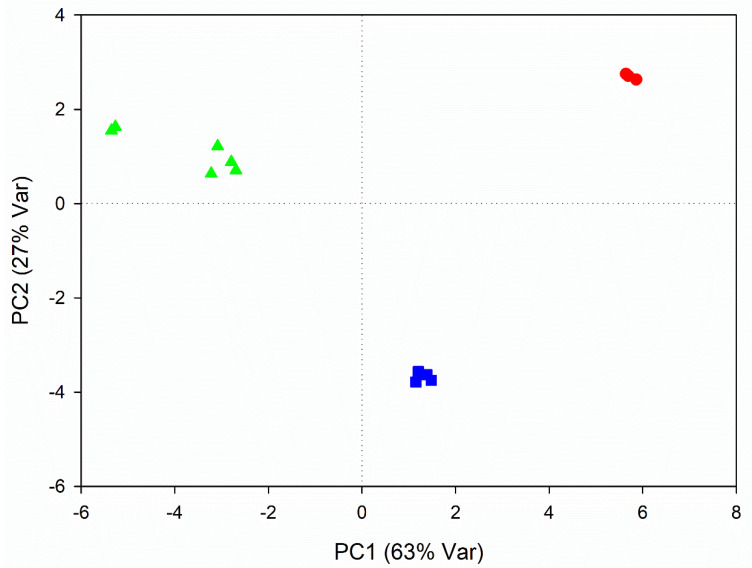
PCA score plot of RB juices from different years. Juices from 2016 are indicated in blue squares, 2017 in red circles and 2018 in green triangles.

**Figure 4 foods-10-01887-f004:**
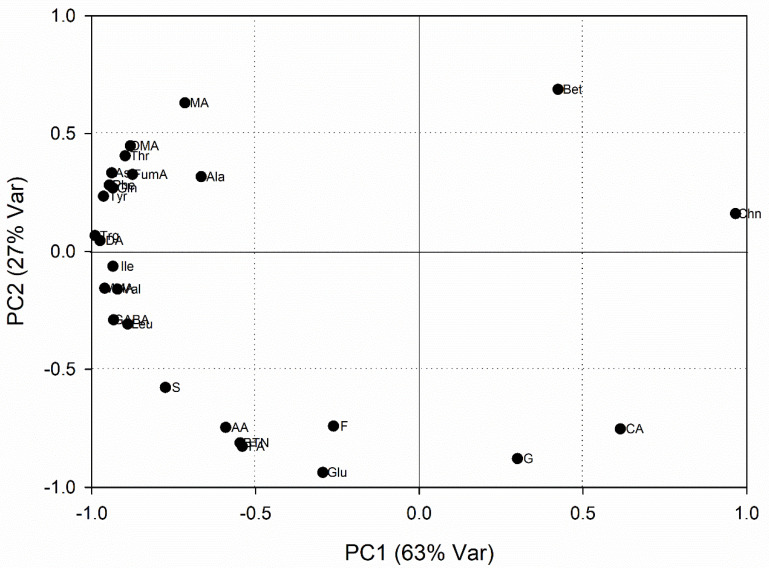
Normalized PCA loading plot of RBs juices produced in the years 2016, 2017 and 2018. Molecule abbreviations are reported in Appendix A.

**Table 1 foods-10-01887-t001:** Comparison among molecule concentrations considered significant by PCA analysis from RBs harvested in July, August and September. The differences in the composition were monitored from July to August and from August to September §: significant differences between July and August; †: between August and September; * *p* < 0.05; ** *p* < 0.01.

Metabolite	(mg/100 g)		
	July	August	September
Glucose	20.0 ± 5.0	42.1 ± 8.6 § **	118 ± 50 † *
Myo-Inositol	0.25 ± 0.27	0.50 ± 0.43 § *	9.72 ± 1.98 † **
Dopamine	7.67 ± 0.18	5.60 ± 0.86 § *	5.81 ± 2.45
Glutamine	245 ± 27	260 ± 34	250 ± 48
Malonate	3.05 ± 0.31	3.19 ± 0.37	3.46 ± 0.53
Fructose	32.2 ± 5.4	23.4 ± 6.5 §*	9.38 ± 0.45 † *
Aspartate	21.8 ± 3.1	14.5 ± 2.8 §*	8.3 ± 1.9 † *
Hydroxybenzoate	1.20 ± 0.15	0.97 ± 0.22	0.48 ± 0.15 † *
Citrate	57.8 ± 6.7	52.2 ± 3.1	40.9 ± 6.7
GABA	10.06 ± 0.81	9.80 ± 0.92	9.4 ± 1.3
Betanin	56.7 ± 6.6	37.9 ± 4.5 § *	23.0 ± 3.1 † *
Choline	3.14 ± 0.22	2.29 ± 0.17 § **	1.63 ± 0.22 † *
Sucrose	3196 ± 267	2514 ± 228 § *	2049 ± 144 † *

**Table 2 foods-10-01887-t002:** Comparison among molecule concentrations considered significant by PCA analysis from RB juice produced in years 2016, 2017 and 2018 with ANOVA significance. The same letter (a, b and c) indicates the variables which are not different among the years according to the Holm–Sidak multiple comparison test.

Metabolite	(mg/100 mL)		
	2016	2017	2018
Leucine	31.7 ± 4.1 ^a^	9.35 ± 0.79 ^b^	31.2 ± 3.4 ^a^
Isoleucine	19.4 ± 2.1 ^a^	6.48 ± 0.54 ^b^	22.5 ± 2.1 ^a^
Valine	16.6 ± 1.8 ^a^	6.70 ± 0.55 ^b^	16.3 ± 1.4 ^a^
Threonine	6.38 ± 0.63 ^a^	4.09 ± 0.30 ^a^	24.5 ± 2.9 ^b^
Alanine	47.5 ± 4.6 ^a^	31.5 ± 2.5 ^a^	44.8 ± 3.7 ^a^
Glutamate	177 ± 20 ^a^	24.3 ± 1.9 ^b^	82.8 ± 6.6 ^c^
Glutamine	233 ± 14 ^a^	134 ± 10 ^b^	291 ± 13 ^c^
Asparagine	29.8 ± 2.8 ^a^	16.4 ± 1.5 ^a^	57.2 ± 4.4 ^b^
GABA	39.8 ± 3.5 ^a^	12.5 ± 1.0 ^b^	40.0 ± 3.5 ^a^
Tyrosine	6.53 ± 0.56 ^a^	3.48 ± 0.24 ^b^	8.94 ± 0.74 ^c^
Phenylalanine	1.83 ± 0.17 ^a^	0.90 ± 0.07 ^b^	3.47 ± 0.30 ^c^
Fructose	497 ± 109 ^a^	111.1 ± 6.5 ^b^	266 ± 28 ^c^
Glucose	1185 ± 113 ^a^	560 ± 42 ^b^	647 ± 46 ^b^
Sucrose	14482 ± 1453 ^a^	6725 ± 538 ^b^	12005 ± 1077 ^a^
Citrate	147 ± 13 ^a^	54.0 ± 3.5 ^b^	17.9 ± 2.7 ^c^
Malonate	5.54 ± 0.48 ^a^	2.09 ± 0.31 ^b^	5.82 ± 0.45 ^c^
Malate	137 ± 12 ^a^	114.0 ± 8.8 ^b^	189 ± 14 ^c^
Fumarate	0.44 ± 0.15 ^a^	0.25 ± 0.04 ^a^	0.91 ± 0.07 ^b^
Formate	4.45 ± 0.43 ^a^	0.80 ± 0.08 ^b^	2.77 ± 0.23 ^c^
Acetamide	5.78 ± 0.94 ^a^	0.12 ± 0.06 ^b^	3.60 ± 0.47 ^a^
Dimethylamine	5.45 ± 0.77 ^a^	10.22 ± 0.63 ^b^	130.5 ± 9.3 ^c^
Choline	23.3 ± 2.6 ^a^	30.4 ± 2.4 ^a^	6.15 ± 0.48 ^b^
Betaine	297 ± 28 ^a^	248 ± 20 ^a^	260 ± 22 ^a^
Dopamine	19.4 ± 1.5 ^a^	4.16 ± 0.24 ^b^	31.5 ± 2.9 ^c^
Betanin	193 ± 30 ^a^	17.8 ± 1.4 ^b^	115.0 ± 8.8 ^c^
Trigonelline	1.42 ± 0.14 ^a^	0.51 ± 0.14 ^b^	1.97 ± 0.16 ^a^

**Table 3 foods-10-01887-t003:** Climatic conditions in Fucino area during the examined years of RB juice production.

Year	Climatic Parameters	Month			
		June	July	August	September
2016	Average Min. Temperature (°C)	12.8	15.7	14.2	12.0
	N° of days with Temperature < 11 °C	2	1	1	3
	Average Max. Temperature (°C)	29.0	32.8	35.4	25.5
	Total mm of precipitations	39.2	44.4	86.4	75.8
2017	Average Min. Temperature (°C)	14.5	15.2	15.7	11.5
	N° of days with Temperature < 11 °C	1	1	1	4
	Average Max. Temperature (°C)	32.5	34.4	35.8	26.1
	Total mm of precipitations	5.2	15.8	4.6	69.9
2018	Average Min. Temperature (°C)	12.9	15.5	14.1	12.0
	N° of days with Temperature < 11 °C	6	2	5	8
	Average Max. Temperature (°C)	29.2	33.5	32.3	28.3
	Total mm of precipitations	41.4	15.8	133.6	19.2

**Table 4 foods-10-01887-t004:** Yearly nitrate content in RB juices.

Year	Nitrate Content (ppm)
2016	3263 ± 262
2017	2883 ± 47
2018	1709 ± 67

## Data Availability

Not applicable.

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
