# Peer review of "Red Beetroot’s NMR-Based Metabolomics: Phytochemical Profile Related to Development Time and Production Year"

_foods, 2021, doi:10.3390/foods10081887_

Round 1

Reviewer 1 Report

Dear Authors,

thank you for your interesting manuscript.

I think that some parts of the manuscript should be more focused and without general information. The manuscript is reminiscent of a textbook in places (e.g. l. 42-49; l. 50-60). Discussion is very poor; l. 212 – 223 is not discussion. In general, references need to be up to date - lack from 2021. More than 40 papers related to beetroot and phytochemical profile were published in 2020-2021.

The manuscript is not balanced or does not match the title. In particular, it focuses more on metabolites than on mechanisms and utilization (due to harvest time, etc.). Beets are nitrate accumulators and there is an interesting interaction between nitrate nutrition and salt stress tolerance and Na + other metabolites accumulation, these points are entirely missing in this manuscript. Do you have any explanation for the location of your experimental field – soil properties + Mehlich III? Surely, was the same field used each year for beets growing?

Add more information on absorption pathways (vacuolar) of plants (beets). I recommend adding: 10.3389/fnut.2021.660150; 10.1016/j.sjbs.2020.10.048; 10.3390/antiox9100960; 10.3390/molecules25225395; 10.3390/molecules26113147

The discussion section needs to be revised. The arguments require a clearer and more precise presentation. Understanding of plant uptake mechanisms is limited because it is restricted to papers that take a specific view and deliberately ignore alternatives, and do not present a balanced view of the evidence.

Figures are unreadable - split and enlarge; significance is strange (Fig. 1 CCA or DCA or RDA?). Fig. 2 – significance? HSD missing. Fig. 3 – illegible, improve years for readers. Fig. 4 – same as Fig. 3 no significance almost, no explanation about PCA analysis. Did you try to compare Control and variants? Or percentage of Control?

Table 1. big variance in content – do you have any explanation? Did you try another statistical test? HSD Tukey? Control variant is July?

Table 2 – again, July means control variant?

Table 3 – can be omitted

Table 4 is important due to nitrate beet accumulator ... improve explanation on l. 278 – 280.

Author Response

Dear Authors,

thank you for your interesting manuscript.

I think that some parts of the manuscript should be more focused and without general information. The manuscript is reminiscent of a textbook in places (e.g. l. 42-49; l. 50-60).

The authors thank the Reviewer for this comment, but at the same time they would like to keep this information in the Introduction as not every reader would be aware on the functional applications of red beetroot in exercise and cardiovascular diseases.

Discussion is very poor; l. 212 – 223 is not discussion. In general, references need to be up to date - lack from 2021. More than 40 papers related to beetroot and phytochemical profile were published in 2020-2021.

The authors kindly thank the reviewer for the suggestion, that portion of the discussion has been modified and bibliography was updated accordingly.

The manuscript is not balanced or does not match the title. In particular, it focuses more on metabolites than on mechanisms and utilization (due to harvest time, etc.). Beets are nitrate accumulators and there is an interesting interaction between nitrate nutrition and salt stress tolerance and Na + other metabolites accumulation, these points are entirely missing in this manuscript.

The authors thank the Reviewer for the opportunity to clarify the aim of this study, which is focused on the changes occurring to the phytochemical profile of red beetroots according to the development time as well as seasonality. In agreement with referee’s suggestion the title has been changed while the implication regarding the harvest time related to different utilization was specified in the discussion (lines 232-238).

Regarding the interaction between soil composition and red beetroots development, given that the soil composition is largely the same during this study (see the next comments and Table S1 in supporting information), the main factors influencing the beet composition are the root development time and the climatic conditions, and both parameters are object of discussion. 

We agree that nitrate nutrition is important for the plant metabolism, particularly for the red beetroot. However, the study was focalized toward the phytochemical profile and the nitrate content showed in Table 4 only represented supporting information.

Do you have any explanation for the location of your experimental field – soil properties + Mehlich III? Surely, was the same field used each year for beets growing?

We thank Reviewer for the opportunity to specify this aspect of the study. The samples come from experimental fields located in the Fucino plain. At the beginning of the experimentation, parcels of land with similar characteristics were identified to be used for the sowing in the different years as, given the type of cultivation, it is customary to use crop rotation to eliminate the problem of soil depletion. The information on the physico-chemical characteristics of the soil has been added to the supporting information Table S1.

Add more information on absorption pathways (vacuolar) of plants (beets). I recommend adding: 10.3389/fnut.2021.660150; 10.1016/j.sjbs.2020.10.048; 10.3390/antiox9100960; 10.3390/molecules25225395; 10.3390/molecules26113147

The authors kindly thank the reviewer for the suggestions, and bibliography was updated accordingly.

The discussion section needs to be revised. The arguments require a clearer and more precise presentation. Understanding of plant uptake mechanisms is limited because it is restricted to papers that take a specific view and deliberately ignore alternatives, and do not present a balanced view of the evidence.

The discussion section has been modified to better distinguish the influence of root development from the one of the seasonal climatic changes. Again, regarding the plant uptake mechanisms, the physiological aspects suggested by reviewer were outside the scope of this work. Our data and the reported bibliography all agree with our hypothesis that the beetroot metabolism was influenced by the external temperature.

Figures are unreadable - split and enlarge; significance is strange (Fig. 1 CCA or DCA or RDA?). Fig. 2 – significance? HSD missing. Fig. 3 – illegible, improve years for readers. Fig. 4 – same as Fig. 3 no significance almost, no explanation about PCA analysis. Did you try to compare Control and variants? Or percentage of Control?

As suggested by the referee, the figures and their respective captions have been modified to make them clearer.

In this study, a single variety of RB was considered (Pablo cultivar). Moreover, the samples considered were collected in different months (July, August and September) and the juices compared for years of production (2016, 2017, 2018). Therefore, as the study was designed, “control” and “variant” groups do not exist.

The PCA analysis, on the other hand, allows to observe spontaneous groupings of samples based on the covariances of the variables. Figures 1 and 3 show the score plots that indicate spontaneous groupings of the samples respectively as a function of the time of collection (Figure 1) and the year of production (Figure 3). Figures 2 and 4, on the other hand, are the loading plots, respectively as a function of the collection time (Figure 2) and the year of production (Figure 4), which show the correlation value among the variables for the components represented in the scores plot. For each PCA analysis, the Pearson correlation values ​​were defined as statistically significant when their value exceeded the threshold value defined on the basis of the number of samples (lines 119-121; 156-159). No further statistical test was carried out on the PCAs, but the variables found to be significant were subsequently subjected to the univariate ANOVA analysis (Tables 1 and 2), as reported in the materials and methods section in the supporting information.

Table 1. big variance in content – do you have any explanation? Did you try another statistical test? HSD Tukey? Control variant is July?

All variables passed the normality test, therefore the corresponding parametric test was applied to differentiate among the sub-groups (Holm Sidak) as reported in Material and Methods. Again, since in this section of the study we are observing a temporal evolution of the same species, it is not correct to speak of “control” or “variant” groups.

In particular, the statistical differences we considered are the ones between consecutive months of development, i.e. from July to August and from August to September. The caption of the table has been improved to clarify this approach.

Table 2 – again, July means control variant?

In this whole experimentation, the same red beetroot cultivar (Pablo) was employed as reported in Material and Methods, so there are no “control” or “variant” categories. In Table 2 the concentrations of RB juices produced in three different years (2016, 2017 and 2018) are reported and the differences among the groups was assessed by one-way ANOVA followed by Holm Sidak multiple comparison tests. The caption of the table has been improved to clarify this approach.

Table 3 – can be omitted

The authors consider that table 3 is useful to readers for a better understanding of the climatic phenomena under consideration. Since the information contained in the table is discussed in the text, we prefer it to be present in the main text and not moved to supplementary.

Table 4 is important due to nitrate beet accumulator ... improve explanation on l. 278 – 280.

The authors thank the reviewer for this remark, yet we consider that this section of discussion is adequate for the objective of the study, which was not designed to precisely evaluate the nitrate accumulation. Therefore, we do not have sufficient data to formulate solid hypothesis regarding this phenomenon, but only to observe a reduction in the overall nitrate content in RB juices in the year 2018 and to link this to the number of days with a minimum temperature below 11°C. The cited literature corroborates this hypothesis, but the authors preferred to not further emphasize this aspect of the study.

Reviewer 2 Report

The Authors present an interesting work in 1H-NMR-based metabolomic analysis on Red beetroot. A total of 28 roots were harvested and 15 juice samples were prepared in a mixture of MeOD/D2O. Classical NMR experiment were recorded. Data were qualitatively and quantitatively analyzed and statistical analysis were performed and metabolites variation identified.

I have few questions:

- I regret the weak number of samples used in this study. Did it be impossible to obtain a minimum of 10 samples by group to perform stable statistical analysis?

- We know that the control of the pH is very important in NMR, to limit the chemical shift variation. The samples were prepared in D2O/MeOD. The Authors never indicated information about pH or chemical variation. Are the spectra perfectly superimposed?

- How do you know that protons are completely relaxed with 6.55s of recycle delay?

- Which pulse do you use for your NMR experiment to obtain quantitative data?

- We need information about the treatment of the NMR spectra, because it’s a crucial step to obtain a correct quantification of each metabolite.

- Please give more information about the normalization procedure of your data set.

- Could you explain me, how you created the data matrix before to perform PCA or ANOVA analysis.

- Why the authors don’t use supervised multivariate analysis to complete their statistical approach?

- Please, in the table 1 and 2, homogenized the SD values. For example, 244.99 +/- 26.86 for glutamine. I think 245 +/- 27 is enough.

- I found some mistake in table 1 :

       - Myo-Inositol 0.25 ± 0.27 in july and 0.50 ± 0.43 in august. it is significant, strange for me, because for Hydroxybenzoate 1.20 ± 0.15 in july and 0.97 ± 0.22 in august, here, it is not significant. The difference is more or less the same (0.25 for Myo and 0.23 for Hydro), the SD is higher for Myo. So it's not possible.

       - For Hydroxybenzoate again. is it not significant between August and September, but you have 0.97 ± 0.22 and 0.48 ± 0.15, are you sure ?

       - Same problème with citrate

       - For Myo-Inositol, only one square for the difference between 0.50 ± 0.43 and 9.72 ± 1.98 that indicate a pvalue betwwen 0.05 and 0.01 ??

- For Table 2, can you explain a and b ??

- Please check the result of your statistical analysis.

In table S1 : for Citrate, it's not 2 doublet, but an AB system. The same for Malate, for the CH2 is an ABd. Please checck.

Author Response

Comments and Suggestions for Authors

The Authors present an interesting work in 1H-NMR-based metabolomic analysis on Red beetroot. A total of 28 roots were harvested and 15 juice samples were prepared in a mixture of MeOD/D2O. Classical NMR experiment were recorded. Data were qualitatively and quantitatively analyzed and statistical analysis were performed and metabolites variation identified.

I have few questions:

- I regret the weak number of samples used in this study. Did it be impossible to obtain a minimum of 10 samples by group to perform stable statistical analysis?

We considered in the study only the roots which, at the same development time, had comparable dimensions in order to avoid adding further variability due to a different content of fiber and water.

- We know that the control of the pH is very important in NMR, to limit the chemical shift variation. The samples were prepared in D2O/MeOD. The Authors never indicated information about pH or chemical variation. Are the spectra perfectly superimposed?

There were small differences in the acid content among the samples but overall, there were no appreciable differences in the chemical shift values​. In our laboratory practice, the spectra are processed and integrated individually and manually, as reported in the materials and methods section in the supporting information, and so there is no superimposition procedure.

Moreover, in our experience, the use of buffers creates further inconvenience due to the increase in the concentration of salts, and therefore of ionic strength which affects both the bandwidth and the goodness of the static magnetic field.

- How do you know that protons are completely relaxed with 6.55s of recycle delay?

The total repetition time of each spectrum is 15s, of which 5.45s of acquisition, 6.55s of relaxation delay, and 2s of solvent suppression. Experiments not shown allowed to optimize these parameters ​​to ensure complete relaxation of the signals. The details were added to the materials and methods section in the supporting information.

- Which pulse do you use for your NMR experiment to obtain quantitative data?

We employed 90° pulses as reported in the materials and methods section in the supporting information.

- We need information about the treatment of the NMR spectra, because it’s a crucial step to obtain a correct quantification of each metabolite.

The required information was added to the materials and methods section in the supporting information.

- Please give more information about the normalization procedure of your data set.

The required information was added to the materials and methods section in the supporting information.

- Could you explain me, how you created the data matrix before to perform PCA or ANOVA analysis.

The ACD software provides, for each spectrum, a table containing the chemical shifts and the integral values ​​for all the selected resonances. These tables are subsequently inserted in an electronic spreadsheet obtaining a matrix having on the columns the values ​​of the integrals of a single band of each analyte (non-redundant matrix) and on the rows the single samples. Subsequently, the values ​​of the integrals are normalized by the number of hydrogens that generates the resonance and, by comparing the integral thus normalized with that of the internal standard (TSP, final concentration 2mM), a matrix of concentrations is obtained. At this point, the values ​​expressed in mg are further normalized by fresh root weight or by volume of juice. The table is completed by adding a column indicating the category of each sample, functional to the subsequent ANOVA analysis.

- Why the authors don’t use supervised multivariate analysis to complete their statistical approach?

In this study, the PCA analyzes of both development time and seasonality showed spontaneous groupings without requiring an a priori indication of the category. In general, spontaneous groupings observed in the unsupervised analysis are able to take into account numerous factors, such as humidity, temperature, etc., which would be more difficult to explain in a supervised analysis focused on a single factor like development time.

Usually, statistical analysis involves an initial unsupervised approach and if it does not provide results that are clearly interpretable, a supervised analysis is carried out. Therefore, following this analysis, we considered it more interesting to perform the univariate analysis, which provides information that can be more easily exported to other areas and other analytical platforms.

- Please, in the table 1 and 2, homogenized the SD values. For example, 244.99 +/- 26.86 for glutamine. I think 245 +/- 27 is enough.

- I found some mistake in table 1 :

       - Myo-Inositol 0.25 ± 0.27 in july and 0.50 ± 0.43 in august. it is significant, strange for me, because for Hydroxybenzoate 1.20 ± 0.15 in july and 0.97 ± 0.22 in august, here, it is not significant. The difference is more or less the same (0.25 for Myo and 0.23 for Hydro), the SD is higher for Myo. So it's not possible.

       - For Hydroxybenzoate again. is it not significant between August and September, but you have 0.97 ± 0.22 and 0.48 ± 0.15, are you sure ?

       - Same problème with citrate

       - For Myo-Inositol, only one square for the difference between 0.50 ± 0.43 and 9.72 ± 1.98 that indicate a pvalue betwwen 0.05 and 0.01 ??

- For Table 2, can you explain a and b ??

- Please check the result of your statistical analysis.

We thank the reviewer for the observations. The statistical analysis was thoroughly checked and the Tables 1 and 2, as well as their captions, were modified accordingly.

In table S1 : for Citrate, it's not 2 doublet, but an AB system. The same for Malate, for the CH2 is an ABd. Please checck.

We thank the reviewer for the observation. The Table S2 (former Table S1) was thoroughly checked and modified according to the reviewer’s suggestion.

Round 2

Reviewer 1 Report

The authors have addressed all the comments and significantly improved the manuscript. In its present form it is ready for acceptance.

This manuscript is a resubmission of an earlier submission. The following is a list of the peer review reports and author responses from that submission.